# Cross-Talks between Raf Kinase Inhibitor Protein and Programmed Cell Death Ligand 1 Expressions in Cancer: Role in Immune Evasion and Therapeutic Implications

**DOI:** 10.3390/cells13100864

**Published:** 2024-05-17

**Authors:** Mai Ho, Benjamin Bonavida

**Affiliations:** Department of Microbiology, Immunology & Molecular Genetics, David Geffen School of Medicine, Jonsson Comprehensive Cancer, University of California, Los Angeles, CA 90095, USA; maiho123@ucla.edu

**Keywords:** immune evasion, cancer, RKIP, PD-L1, cross-talk, targeted therapy

## Abstract

Innovations in cancer immunotherapy have resulted in the development of several novel immunotherapeutic strategies that can disrupt immunosuppression. One key advancement lies in immune checkpoint inhibitors (ICIs), which have shown significant clinical efficacy and increased survival rates in patients with various therapy-resistant cancers. This immune intervention consists of monoclonal antibodies directed against inhibitory receptors (e.g., PD-1) on cytotoxic CD8 T cells or against corresponding ligands (e.g., PD-L1/PD-L2) overexpressed on cancer cells and other cells in the tumor microenvironment (TME). However, not all cancer cells respond—there are still poor clinical responses, immune-related adverse effects, adaptive resistance, and vulnerability to ICIs in a subset of patients with cancer. This challenge showcases the heterogeneity of cancer, emphasizing the existence of additional immunoregulatory mechanisms in many patients. Therefore, it is essential to investigate PD-L1’s interaction with other oncogenic genes and pathways to further advance targeted therapies and address resistance mechanisms. Accordingly, our aim was to investigate the mechanisms governing PD-L1 expression in tumor cells, given its correlation with immune evasion, to uncover novel mechanisms for decreasing PD-L1 expression and restoring anti-tumor immune responses. Numerous studies have demonstrated that the upregulation of Raf Kinase Inhibitor Protein (RKIP) in many cancers contributes to the suppression of key hyperactive pathways observed in malignant cells, alongside its broadening involvement in immune responses and the modulation of the TME. We, therefore, hypothesized that the role of PD-L1 in cancer immune surveillance may be inversely correlated with the low expression level of the tumor suppressor Raf Kinase Inhibitor Protein (RKIP) expression in cancer cells. This hypothesis was investigated and we found several signaling cross-talk pathways between the regulations of both RKIP and PD-L1 expressions. These pathways and regulatory factors include the MAPK and JAK/STAT pathways, GSK3β, cytokines IFN-γ and IL-1β, Sox2, and transcription factors YY1 and NFκB. The pathways that upregulated PD-L1 were inhibitory for RKIP expression and vice versa. Bioinformatic analyses in various human cancers demonstrated the inverse relationship between PD-L1 and RKIP expressions and their prognostic roles. Therefore, we suspect that the direct upregulation of RKIP and/or the use of targeted RKIP inducers in combination with ICIs could result in a more targeted anti-tumor immune response—addressing the therapeutic challenges related to PD-1/PD-L1 monotherapy alone.

## 1. Introduction

The immune system contributes significantly to safeguarding the body against infections, foreign antigens and grafts, and cancer through various mechanisms. Integral to this system, for cancer, are the innate and adaptive immune responses. The innate cell-mediated response activates non-specific immune responses in the presence of foreign antigens and occurs independently of specific recognition [1,2]. This system hosts primarily a population of natural killer (NK) cells, Gamma delta (γδ) T cells, and dendritic cells and macrophages that can affect tumor cells’ viability. They can also process cancer cell antigens and present these antigens on major histocompatibility complexes’ classes I and II (MHC I; MHCII) molecules to both T and B cells [1,3,4]. In contrast, the adaptive immune response can utilize both B cells and T cells to induce a specific antibody response and a cytotoxic immune response, respectively [5]. Despite these above defenses, cancer cells can still circumvent the immune response by various mechanisms, such as downregulating MHC I molecules to silence antigen presentation [5], or recruiting M2-like tumor-associated macrophages (TAMS) to promote tumorigenesis [6]. Furthermore, cancer cells can acquire a resistant phenotype or create an immunosuppressive TME that decreases the anti-tumor efficacy of T cells, which can lead to primary drug resistance and decreased immune surveillance [4,7]. The TME is complex and harbors many immune suppressive cells such as regulatory T cells (Tregs), TAMs, myeloid-derived suppressor cells (MDSCs), etc., that inhibit the CD8+ T cells anti-tumor cytotoxicity and cytokine production [8,9]. Moreover, interactions between tumor cells and the TME can induce the upregulation of checkpoint receptors (e.g., PD-1, LAG3, TIM3, etc.) and their corresponding ligands [10]. This can lead to the inactivation and exhaustion of CD8+ T cells, resulting in immune evasion [10,11].

Thus, over the past decades, many cancer immunotherapies have made significant advances under clinical applications. Examples include the chimeric antigen receptor (CAR) T cell therapy that has been successfully FDA approved for CD19 lymphomas [12] and BCMA in multiple myelomas [13]. In addition, genetically engineered T cells known as tumor-infiltrating lymphocytes (TIL ACT) can cause tumor regression in several types of cancers including melanoma, cervical squamous cell carcinoma, and cholangiocarcinoma through the incorporation of tumor-recognizing receptors [14,15]. However, a more widely used immunotherapy is immune checkpoint inhibitors (ICIs), which are monoclonal antibodies that can block interactions between immune checkpoint proteins and prevent cancer cells from evading immune surveillance [1]. ICIs can target checkpoint receptors using various antibodies capable of recognizing them on both naïve and cytotoxic T cells, as well as against their corresponding ligands [16]. Immune checkpoint receptors are crucial regulators of the anti-cancer immune response and consist of a range of co-inhibitory receptors and their corresponding ligands. Specifically, the discovery of singular receptors was reported, namely, cytotoxic T lymphocyte antigen 4 (CTLA-4) [17] and the subsequent breakthroughs with the programed cell death 1 (PD-1) receptor (CD279) and its ligands PD-L1 (CD274) and PD-L2 (CD273) [18,19]. PD-1 is a cell surface receptor primarily found on tumor-infiltrating cells, regulating T cells and preventing them from attacking healthy cells [20]. PD-L1 is a type I transmembrane protein mainly expressed on antigen-presenting cells and tumor cells, serving as the ligand for PD-1 and activating this coreceptor. When the PD-1/PD-L1 axis is activated, it serves to protect against excessive immune responses and autoimmune reactions [21]. However, tumor cells have the capacity to exploit this mechanism, transmitting signals that upregulate PD-1 on the CD8 T cells, effectively blocking T cell effector functions and permitting immune evasion [5,10,22].

Raf Kinase Inhibitor Protein (RKIP) belongs to the phosphatidylethanolamine-binding protein (PEBP) family. RKIP is a dynamic cytosolic protein widely expressed in many normal human tissues (brain, testis, epididymis, liver, kidney, etc.) [23]. It has important roles in various physiological processes, including cardiac and neurological outputs, spermatogenesis, and membrane biosynthesis [24]. Importantly, RKIP was first reported by Yeung et al., as the first protein that inhibits the Raf/Mek pathway [25]. Since then, RKIP has been reported to be involved in the regulation of many signaling pathways that govern cellular growth, division, migration, and apoptosis [26,27]. As a result, RKIP plays a pivotal role in the pathogenesis and development of many types of cancers, being characterized as a tumor suppressor, an immune modulator [28], a cancer biomarker [29], and a crucial therapeutic target [23]. This growing understanding of RKIP’s molecular roles is instrumental in the development of targeted immunotherapies, offering potential avenues to improve the body’s natural defenses against cancer.

The objective and scope of this review are to examine the various cross-talks between RKIP and PD-L1, focusing on how their expressions correlate in cancer. Investigations into such cross-talks offer new insights into the potential development of future immunotherapeutic interventions that target the potential RKIP-PD-L1 axis to combat cancer’s immune evasion strategies. Below, we present the role of the PD-L1 pathway in cancer immune evasion, the role of RKIP in anti-tumor immune activation, the signaling cross-talks between RKIP and PD-L1 expressions, and targeted therapeutic strategies to avoid immune evasion.

### 1.1. The PD-L1/PD-1 Pathway in Immune Evasion

The interaction between PD-L1 and PD-1 on anti-tumor CD8 T cells results in a signaling process that leads to reduced T cell function and T cell exhaustion as well as reduction in the synthesis of inflammatory cytokines [30]. PD-1 is unique from the CD28 superfamily, although it was initially considered a member of that family due to some structural similarities. While CD28, CTLA-4, and ICOS have Src homology (SH2)-binding motifs and SH3-binding motifs in their cytoplasmic tails, PD-1 lacks these motifs [31]. Instead, the intracellular domain of PD-1 contains distinct phosphorylation sites at its N-terminal and C-terminal amino acid residues, specifically in the immunoreceptor tyrosine-based inhibitory motif (ITIM) and the immunoreceptor tyrosine-based switch motif (ITSM), respectively [32]. These motifs are highly conserved amino acid sequences that can recruit Src homology domains 1 and/or 2 containing phosphatases (SHP1 and SHP2) that can inhibit cellular activation [33]. Additionally, PD-1 exists as a monomer, while CD28, ICOS, and CTLA4 exist as dimers. These differences set PD-1 apart from the CD28 superfamily, with potential implications for its distinct regulatory functions. When PD-1 on T cells interacts with PD-L1 on tumor cells, it undergoes phosphorylation at specific tyrosine residues within its intracellular domain that activates a signaling cascade. Upon binding with PD-L1, the ITSM tyrosine residue (Y248) is phosphorylated, resulting in the recruitment and binding of SHP2 [18,34]. Patsoukis et al. identified a mechanism whereby SHP2 can bridge phosphorylated ITSM-Y248 residues via its N-SH2 and C-SH2 domains, to form a PD-1 dimer in live cells. Interactions between SHP2 and ITSM-Y248 can dephosphorylate key signaling kinases and counteract positive signals (CD-3ζ and ZAP70) that occur through the activation of the T cell receptor (TCR) and CD28 receptors, leading to the suppression of T cell functions and the development of T cell exhaustion [18,35,36]. Furthermore, this process results in the downstream inhibition of various pathways, including the phosphoinositide 3-kinase/protein kinase (PI3K/AKT) and Ras/MEK/ERK/MAPK pathways, which are key regulators of T cell metabolism [35,37,38]. Specifically, within the PI3K/AKT pathway, PD-1 has the capability to inhibit casein kinase 2 (CK2) and downregulate the phosphatase and tensin homolog (PTEN), while increasing phosphatase activity [31]. This downregulation of PTEN is associated with the unchecked cell growth and division often observed in cancer development [39]. Over time, the signaling pathway of PD-1/PD-L1 ultimately results in the reduced activation of transcription factors (TFs) such as activator protein 1 (AP1), nuclear factor of activated T cells (NFAT), and nuclear factor kappa B (NF-κB) (Figure 1). These also counteract with favorable signals that promote T cell activation, proliferation, effector functions, and survival [40]. In the past years, Tang et al. [30], and Wang et al. [41], reported that PD-L1 is upregulated by cytokines such as interferon-γ (IFNγ), tumor necrosis factor alpha (TNFα), and interleukin-6 (IL-6) on APCs in the TME and lymph nodes to inhibit T cell activation. PD-L1 upregulation can also be enhanced by reducing microRNA expressions in tumor cells such as miR-200, miR-34a, miR-152, and miR-424 [42,43]. As a result, through many mechanisms and transcriptional control, cancer cells can upregulate PD-L1/PD-L2, enhancing their ability to suppress T cells and hinder the host’s immune responses.

### 1.2. PD-L1 and Inactivation of CD8+ T Cells

The first indication of the PD-L1/PD-1 pathway’s involvement in immune evasion dates back to 2002, when researchers found that the overexpression of PD-L1 could diminish the cytolytic activity of T cells, thus promoting tumorigenesis and immune evasion [44]. These discoveries prompted further investigations into PD-L1/PD-1 in the inactivation and exhaustion of CD8+ T cells [11]. CD8+ T cells play a vital role in controlling the growth of tumors and responding to chronic viral infections [45]. However, in cancer, CD8+ T cells can become exhausted and they progressively lose their ability to carry out their effector functions, resulting in cell apoptosis and dysfunction [11,46]. A key feature of these exhausted T cells is the elevated and prolonged expression of PD-1. According to Liu et al., this elevated expression is stimulated by tumor-repopulating cells (TRCs) in CD8+ T cells through a transcellular pathway involving kynurenine (Kyn) and the aryl hydrocarbon receptor (AhR) [47]. CD8+ T cells, via the production of IFNγ, stimulate the release of substantial amounts of Kyn from TRCs. This Kyn is then transferred into neighboring CD8+ T cells using transporters SLC7A8 and PAT4. Kyn activates AhR, leading to the upregulation of PD-1 expression. This Kyn-AhR pathway has been validated in both tumor-bearing mice and patients with cancer, and blocking this pathway enhances the effectiveness of adoptive T cell therapy against tumors [47]. Additionally, it is known that PD-1 is also expressed during the naïve-to-effector CD8 T cell transition, not only on exhausted T cells [48,49]. However, Ahn et al. discovered that PD-1 was rapidly upregulated by TCR signaling upon T cell activation even before cell division. Importantly, when they blocked PD-1/PD-L1 interaction during the early phase, it significantly enhanced CD8 T cell effector function and viral clearance. These further demonstrated that PD-1 expression on early activated T cells has an inhibitory role in T cell immunity.

### 1.3. Clinical Implications of the PD-1/PD-L1 Axis

The mechanisms and signal transduction pathways mentioned above demonstrate how PD-L1 expression on tumor cells and binding with PD-1 on CD8 T cells play a significant role in immune evasion and tumor progression. In the context of cancer, the PD-L1/PD-1 axis becomes crucial. Developments of ICIs blocking PD-1 and PD-L1 interactions can restore the functional abilities of CD8+ T cells in the TME, resulting in improved control of viral replication and tumor growth. Therapeutic antibodies targeting PD-L1 include atezolizumab, avelumab, and durvalumab [50]. For PD-1, antibodies such as nivolumab, pembrolizumab, and cemiplimab have been developed, showing promising outcomes in clinical trials for various cancer types [50,51]. This approach helps restore the balance in anti-tumor immune responses and has achieved response rates ranging from 10% to 40% in clinical settings [16]. Currently, atezolizumab, nivolumab, and pembrolizumab have received approval from the FDA for treating multiple cancer types, including melanoma, small cell lung cancer (SCLC), non-small cell lung cancer (NSCLC), renal cell carcinoma (RCC), head and neck squamous cell carcinomas (HNSCCs), classical Hodgkin lymphomas (cHLs), and Merkel cell carcinoma [52].

## 2. RKIP Properties and Immune Activation

RKIP was first isolated from a bovine brain, and termed phosphatidylethanolamine-binding protein 1 (PEBP1) due to its interaction with phosphatidylethanolamine (PE) [53]. More than a decade later, Yeung et al. discovered a breakthrough in the function of the protein, revealing its ability to inhibit Raf-1 in the mitogen-activated protein kinase (MAPK) pathway, earning its RKIP name [25]. Since then, researchers have elucidated RKIP’s role in many signaling cascades beyond MAPK [54,55,56,57], along with a growing number of studies examining the loss of RKIP expression in many types of cancer [29,55,58].

Structurally, RKIP is a small cytosolic protein (23 kDa), with a highly conserved binding pocket [59]. The pocket comprises 16 amino acid residues and can host various nucleotides such as non-lipid organic compounds and phospholipids [56,60]. Through this flexible pocket, RKIP/Raf-1 interaction can exist in discrete states that allosterically regulate functional switching to trigger specific signaling cascades [61]. This allows for RKIP to switch protein partners, as well as change its function. For example, it has been established that phosphorylation at the Ser153 residue by PKC transforms RKIP from a state that binds to Raf (RKIPRaf) to one that binds to G Protein-Coupled Receptor Kinase 2 (GRK2) termed RKIPGRK2 [62]. These two states are modulated through a novel phosphorylation-induced salt bridge theft. In its unphosphorylated form, RKIP possesses a lysine at position 157, engaging in a salt bridge with negatively charged amino acids (D134 and E135) on an adjacent region of the protein. Upon phosphorylation at S153, a strong negative charge is induced and enables the nearby serine to compete with the negatively charged residues in the neighboring salt bridge. This competition results in the phosphorylated serine ‘stealing’ the positively charged lysine and forming a new salt bridge [57,63]. This newly established salt bridge induces the functional alteration in RKIP as previously described. Additionally, researchers reaffirm the existence of a third state, termed RKIPKin, that interacts with kinases responsible for phosphorylating RKIP based on experiments involving the RKIP loop mutant P74L [59,61]. Thus, these multiple discrete states of RKIP provide the molecular basis for its multifunctional role in regulating oncogenesis and function as a tumor suppressor. 

### 2.1. RKIP Signaling Pathways

RKIP was the first identified endogenous inhibitor of the Raf-1-MEK-ERK pathway, a major hyperactive pathway in cancer. It achieved this by employing two mechanisms: (1) binding to the N-region of the Raf-1 kinase domain, thereby inhibiting Ser338 and Tyr340/341 phosphorylation and preventing Raf-1 activation, and (2) dissociating the Raf-1/MEK complex, through competitive inhibition with MEK phosphorylation [64,65]. MAPK signaling plays a significant role in regulating cell proliferation, differentiation, and survival [66]. Thereby, its heightened activation in cancer cells is linked to the initiation of metastasis and resistance to therapeutic interventions [67].

In contrast to the unphosphorylated RKIP form, upregulated phosphorylated RKIP (pRIKP) functions by competitively inhibiting survival signals and apoptosis in cancer cells [68,69]. Following phosphorylation at Ser153, RKIP dissociates from Raf-1 and forms a complex with GRK2, an inhibitor of G protein-coupled receptors (GPCRs) [62,70]. This is carried out indirectly by disrupting upstream activators of Raf-1. Due to the release of Raf-1 from RKIP inhibition, the interaction between phosphorylated RKIP and GRK2 not only augments GPCR activation but also contributes to the hyperactivation of the MAPK pathway [28]. Hence, certain cancer types like colon and gastric cancers and multiple myeloma overexpress pSer153 RKIP, correlating with an unfavorable prognosis [71,72,73]. However, it has been shown that pRKIP can also activate PKA signaling, leading to its emerging role in preventing cardiac failure through the upregulation of β-adrenoceptor/PKA signaling [57].

Another major pathway mediated by RKIP is inhibiting the signaling of the NF–κB pathway through interactions with various upstream kinases including transforming growth factor β-activated kinase 1 (TAK-1), NF-κB-inducing kinase (NIK), and the IκB kinase alpha and beta (IKKα and IKKβ), resulting in the elimination of the IkappaB α (IκBα) phosphorylation and degradation, preventing NF-κB translocation to the nucleus [74]. Through the inhibition of NF–κB, RKIP hinders antiapoptotic genes and facilitates the de-repression of proapoptotic genes through the NF-kB/Snail/Yin Yang 1 (YY1)/RKIP/PTEN dysregulated loop [54,75]. This favors the promotion of cell death and malignant cells’ sensitization to apoptosis by various chemotherapeutic and immunotherapeutic drugs [76]. The inhibition of this pathway also leads to downstream suppression of gene products in the epithelial–mesenchymal transition (EMT), tumor invasion, and cell metastasis [77,78].

RKIP also has activities with other key hallmarks of cancer, such as the inhibition of the signal transducer and activator of transcription 3, STAT3. It can carry out this by blocking upstream kinases such as Janus kinases 1 and 2 (JAK1, JAK2) and inhibiting phosphorylation. JAK1 and JAK2 phosphorylate STAT3 at Tyr705, resulting in the translocation of activated STAT3 dimers to the nucleus. The constitutive activation of STAT3 is linked to tumor invasion, angiogenesis, and cell metastasis [79,80]. Moreover, STAT3 has also emerged as a pivotal factor in inflammation-mediated cancer, metabolism, cancer stem cells (CSCs), and the formation of pre-metastatic niches [81,82].

It is important to emphasize that the role of RKIP extends beyond inhibition within the signaling pathways. Recent research affirms that RKIP, through the modulation of diverse protein–protein interactions, plays a more intricate role in fine-tuning cell signaling [83], autoimmune inflammation [84], and immune resistance [85], serving as a prognostic/diagnostic biomarker [69,86], and regulating cross-talk between various pathways [72]. Recent investigations have also revealed its widening impact on cancer, influencing mitochondrial functions, DNA methylation, chromatin modulation, and the regulation of autophagy [82,87]

### 2.2. Cross-Talks between RKIP and PD-L1

The significance of RKIP in the immune response and its influence on the TME have been expanding. In addition to mediating pathways crucial in the immune response, RKIP also plays a direct role in the infiltration of specific immune cells, such as TAMs [55,88]. It also participates in pro-inflammatory signaling [89], and regulating tumor re-sensitivity to apoptosis [28]. In contrast to the role of PD-L1 in immune evasion, RKIP’s emerging role as an immune modulator, regulating crucial pathways in immune surveillance and suppression, led us to propose that there exists—directly or indirectly—signaling cross-talks between RKIP and PD-L1 expressions in cancer cells. Below, we will present the various cross-talk pathways that we have compiled and we investigated their relationships between RKIP and PD-L1 (Figure 2).

### 2.3. Cross-Talk via the MAPK Pathway

In the MAPK pathway, there is an indirect regulation between RKIP and PD-L1 expressions. Recent investigations by Jha et al. have reported that PD-L1 is regulated by JAK2-STAT3 and MAPK-AP1 signaling pathways [90]. Both of these pathways are also regulated by RKIP, whereby RKIP acts as an endogenous inhibitor [71,83]. The study used human tissue samples from cases of oral squamous cell carcinoma (OSCC), where a heightened expression of PD-L1 was observed in both tumors and cisplatin-resistant SCC4/9 cells. An analysis of PD-L1 mRNA and protein levels revealed a positive correlation with various transcription factors (AP1, STAT3, and NFκB) in tumor samples, which led to the conclusion that JAK2-STAT3/MAPK-AP1 pathways are primary regulators of PD-L1 expression [90]. Additionally, knockdown of PD-L1 alone reduced invasion and drug resistance, whereas the combinational inhibition of PD-L1, MAPK, and JAK-STAT promoted cell death, suggesting further cross-talks among these molecules/pathways [90]. Multiple other studies have also confirmed the MAPK and PD-L1 axis [91,92,93]. In summary, RKIP and PD-L1 can interact with each other via MAPK signaling and various downstream factors.

### 2.4. Cross-Talk via Cytokines IL-1β and IFN-γ

In a recent study by Hirayama et al., they have shown that inhibitors of MAPK signaling blocked the upregulation of PD-L1 by cytokines interleukin-1 beta (IL-1β) and IFN-γ [94]. The study revealed that IL-1β is abundant in macrophages present in the TME of NSCLC, and IL-1β expression alone upregulates PD-L1 expression in certain NSCLC cell lines. Importantly, their findings also showed that the concurrent activation of IFN-γ and IL-1β achieved the highest increase in PD-L1 expression in a significant portion of (NSCLC) cell lines. They concluded that the activation of the MAPK and ERK signaling pathways in this synergistic effect is crucial for PD-L1 upregulation, and that the use of MAPK inhibitors, SB203580 (p38 inhibitor), SP600125 (JNK inhibitor), and U0126 (inhibitor of the ERK kinase MEK), efficiently blocked this response [94]. Although these inhibitors are synthetic, it offers a similar framework to understand RKIP’s potential regulation of PD-L1 as an endogenous inhibitor within the ERK-MEK/MAPK pathways and additional interactions with PD-L1 via cytokines in the TME. In fact, RKIP has been shown to be highly expressed in epithelial cells, where it can prevent the IL-1β-induced stimulation of protein arginine methyltransferase 1 (PRMT1) expression in the context of lung inflammation [95]. RKIP has also been shown to curb IFN-γ synthesis by CD8+ T cells during serial TCR triggering in the systemic inflammatory response syndrome (SIRS) [96]. The inhibition of RKIP, using locostatin (inhibitor of RKIP), also led to a significantly diminished IFN-γ response in SIRS [89,96]. As mentioned earlier, there has also been numerous investigations into the PD-L1/IFN-γ axis, revealing IFN-γ as a major inducer in the expression of PD-L1 [41,42,97,98]. As a result, possible interactions between cytokines IL-1β and IFN-γ with RKIP in the regulation of PD-L1 via MAPK signaling are a key node to consider for potential cross-talks.

### 2.5. Cross-Talk via GSK3β

Glycogen synthase kinase-3β (GSK3β) is a multifunctional serine/threonine kinase that regulates major cellular pathways [99], including downregulating oncogenic pathways such as the Wnt signaling, EMT, and cyclin D1 activation to suppress tumor progression [100,101]. GSK3β is involved in the phosphorylation of PD-L1, via residues T180 and S184 of PD-L1, promoting the ubiquitin-dependent degradation of PD-L1 [42]. Importantly, it has been shown that RKIP can bind to GSK3 proteins and maintain GSK3β protein levels and its active form [72,100]. The depletion of RKIP increased the oxidative-stress-mediated activation of the p38 MAPK pathway, causing the inactivation of GSK3β by phosphorylation at the inhibitory T390 residue [100]. As such, RKIP can positively modulate GSK3β signaling and prevent its degradation. This provides a negative relationship between RKIP and PD-L1 as GSK3β-mediated phosphorylation causes degradation of PD-L1.

### 2.6. Cross-Talk via the Sox2 Oncogene

We posit the existence of an indirect regulatory relationship between RKIP and PD-L1 mediated by the sex-determining region Y-box 2 (SOX2). SOX2 is an oncogene that plays a pivotal role in cancer cell progression and resistance against various therapeutic interventions. Its overexpression has been associated with the control of key malignant cell features such as proliferation, migration, invasion, metastasis, EMT, sphere and colony formation, tumor initiation, and resistance to apoptosis [102].

In the context of PD-L1 regulation, studies have indicated that the depletion of SOX2 results in reduced PD-L1 expression. Zhong et al. conducted in vitro and in vivo assays, revealing a SOX2 binding site in the promoter region of PD-L1. This binding was confirmed through the direct interaction of SOX2 with the PD-L1 promoter via the consensus SOX2 motif [103]. Subsequent investigations demonstrated that SOX2 promotes the transcriptional activity of the PD-L1 promoter region through this motif [103]. Regarding RKIP’s interaction with SOX2, Cho et al. reported that RKIP induces the nuclear export of SOX2, resulting in a reduction in SOX2 expression [104]. This process is proposed to occur through the inhibition of ERK by RKIP. Consequently, an inverse relationship between RKIP and PD-L1 is proposed, which is mediated through SOX2. Through the modulation of the SOX2 gene, the upregulation of RKIP leads to a decrease in SOX2 expression, ultimately resulting in reduced PD-L1 expression. Additionally, studies have substantiated the existence of cross-talks between RKIP and various cancer stem cell factors including SOX2 [105,106].

### 2.7. Cross-Talk via YY1 and NFκB

Ying Yang (YY1) serves as a pivotal transcription factor in tumor development, with its overexpression strongly correlating with various malignant processes such as cancer metastasis, EMT transition, drug resistance, and unfavorable prognoses [107]. Notably, a complex interplay exists among RKIP, YY1, and NFκB, via the dysregulated NFκB/Snail/YY1/RKIP/PTEN resistance-driver loop. Within this loop, increased RKIP expression suppresses NF-κB, which inhibits YY1 via Snail and induces PTEN expression to suppress PI3K/AKT [54,75,108]. This is highly significant, as YY1 has been shown to promote PD-L1 expression through various pathways [109]. Specifically, YY1 can act as a negative regulator of p53, which inhibits the expression of miR-34a downstream of p53, leading to an increase in PD-L1 expression by binding to the PD-L1 3′UTR [43]. YY1 has also been shown to activate the PI3K/AKT pathway and enhance PD-L1 expression by suppressing PTEN through p53 [43,109]. Furthermore, NFκB has also been reported to directly induce PD-L1 gene transcription by binding to its promoter, validated by ChIP-PCR and luciferase assays [110,111,112]. In addition to the transcriptional role NFκB plays, its activation was also shown to be critical in PD-L1 regulation in multiple cancer lines including melanoma [113], meningioma [114], and gastric cancer [115]. Because of these relationships, promoting RKIP could potentially diminish both YY1 and NFκB expressions, thereby increasing PTEN expression and preventing a further stimulation of the PI3K/AKT pathway and PD-L1 expression.

### 2.8. Analyses of the Correlation between RKIP and PD-L1 Expressions in Human Cancers by Bioinformatics

To further investigate the cross-talks between RKIP and PD-L1, and explore their expressions within different types of cancer, we utilized multiple databases and bioinformatics programs to investigate such cross-talks.

Using the TISIDB (an integrated repository portal for tumor–immune-system interactions) [116], higher RKIP expression corresponded to a significant lower PD-L1 expression in 12 different cancer types. Specifically, the cancer types are colon adenocarcinoma (COAD), stomach adenocarcinoma (STAD), uterine corpus endometrial carcinoma (UCEC), head and neck squamous cell carcinoma (HNSC), Liver Hepatocellular Carcinoma (LIHC), Skin Cutaneous Melanoma (SKCM), Mesothelioma (MESO), thyroid carcinoma (THCA), Sarcoma (SARC), rectum adenocarcinoma (READ), prostate adenocarcinoma (PRAD), and bladder urothelial carcinoma (BLCA) (Figure 3). In all of these cancer types, the Spearman correlation (rho) was negative with significant *p*-values that reached statistical significance below the threshold of 0.01 (Figure 3). As mentioned earlier, cancer cells evade tumor surveillance by upregulating PD-L1, so the negative relationship between PD-L1 and RKIP provides further evidence of cross-talk in these gene products. Given RKIP’s inhibitory roles in tumorigenesis and immune regulation, this provides a potential basis for the RKIP-mediated suppression of PD-L1 in malignant cells. 

Using the gene expression profiling interactive analysis (GEPIA) [117], which derives molecular data from the Cancer Genome Atlas Program (TCGA), we found a significant negative regulation (*p* < 0.05) between RKIP and PD-L1 in 8 out of 34 cancer types provided, namely BLCA, BRCA, COAD, HNSC, cervical squamous cell carcinoma (CESC), esophageal carcinoma (ESCA), LUAD, and thyroid carcinoma (THCA) (Figure 4A). Meanwhile, Pheochromocytoma and Paraganglioma (PCPG), ovarian serous cystadenocarcinoma (OV), pancreatic adenocarcinoma (PAAD), and Uveal Melanoma (UVM) showed a positive correlation between RKIP and PD-L1 (Figure 4B). Interestingly, part of the data show positive relationships between PD-L1 and RKIP in four different types of cancer. This could be because various tumor types are heterogeneous in their makeup and different signaling pathways can influence the overall cross-talks and correlations. Another plausible possibility is that in some cancers, the expression of RKIP is in its inactivated form, namely pRKIP [68,69]. In this state, RKIP expression will no longer lead to a negative correlation. The analyses in Figure 4A,B were based on the expression of RKIP and did not discriminate between active and inactive forms of RKIP.

To further explore its prognostic roles in different types of cancer, we also looked at the survival prognosis with RKIP. Using TISIDB, we observed that higher expression levels of RKIP showed longer survival rates for CESC, KIRC, KIRP, LGG, LUAD, Mesothelioma (MESO), PAAD, and UVM (Figure 5).

Next, we also looked at dysregulation in the expression levels of both PD-L1 and RKIP in both tumor and normal samples of 31 different tumors. The GEPIA analysis revealed that PD-L1 expression levels were significantly dysregulated in Lymphoid Neoplasm Diffuse Large B cell Lymphoma (DLBC), LUAD, lung squamous cell carcinoma (LUSC), Thymoma (THYM), and Uterine Carcinosarcoma (UCS) (Figure 6A). The same analysis was performed with RKIP, where RKIP expression showed similar trends of dysregulation as PD-L1 in DBLC and THYM. It was also dysregulated in Sarcoma (SARC), PCPG, KIRH, and CHOL (Figure 6B). These findings imply the potential of shared immunoregulatory mechanisms between RKIP and PD-L1 across these cancers, which may contribute to similar patterns of dysregulated expression.

Finally, we also looked at RKIP and PD-L1 at the transcriptional level, exploring what transcription factors bind to downstream or upstream regulatory sequences. We employed the motifmap-RNA database [118,119,120], which provides a genome-wide map of regulatory motif sites. We have found five similar motifs that allow for the same transcription factors (TFs) to bind between PD-L1 and RKIP. We have highlighted those TFs, as well as the locations, corresponding binding scores (BLS and BBLS), and Z-scores (Appendix A). Most notably, HMG IY may function as a promoter-specific accessory factor for NF-kappa B transcriptional activity [121], while the LEF1 has shown an important regulation in Wnt signaling [122]. Focusing on these TFs may reveal important implications in the regulation of cancer invasion and metastasis at the transcriptional level between PD-L1 and RKIP.

### 2.9. Potential Therapeutic Strategies Targeting RKIP and PD-L1

#### RKIP Inducers

RKIP upregulation can be induced through several synthetic, semi-synthetic, and natural drugs, as well as specific proteins and microRNAs. Notably, studies have investigated the effects of natural agents such as epigallocatechin gallate (EGCG) and ginseng extract (*Panax quinquefolius* L.) [123,124]. EGCG, a polyphenol derived from green tea, is known for its ability to inhibit NF-κB activity in various human malignancies [125]. In a study conducted by Kim et al., compelling results indicated that EGCG also upregulates RKIP expression by modulating histone deacetylation, leading to the inhibition of Snail expression and decreased NF-κB nuclear translocation in the AsPC-1 human pancreatic adenocarcinoma cell line [124]. Additionally, the administration of ginseng extract has been correlated with a significant increase in RKIP mRNA and protein expression. This is accompanied by a concurrent decrease in phospho-ERK1/2 and -MEK1/2 levels, as well as pRaf-1 in breast carcinoma cells [123]. These findings suggest that ginseng may effectively inhibit cancer cell proliferation by upregulating RKIP expression, leading to the suppression of the MAPK pathway.

In the past years, several semi-synthetic drugs have also been shown to upregulate RKIP expression. Specifically, Rituximab [126], Didymin [127], and Dihydroartemisinin (DHA) are included [128]. DHA is derived from artemisinin, a globally recognized anti-malaria drug. It has also demonstrated the capacity to hinder tumor growth while demonstrating low toxicity to normal cells [129]. In a study by Hu et al., they showed that DHA can induce apoptosis of cervical cancer cells via the upregulation of RKIP in HeLa cells. Their findings also highlighted a significant inhibition of tumor growth in xenografted mice bearing Hela or Caski tumors [128].

Didymin, a citrus-derived natural compound, has shown to induce apoptosis in neuroblastoma by upregulating RKIP and inhibiting N-Myc [127]. Furthermore, in the context of hepatic injury, Didymin has been observed to enhance RKIP expression by inhibiting the MAPK and NF-κB signaling pathways, thereby influencing inflammatory responses [130].

In a recent study by Cho et al., they investigated a novel chemical, Nf18001, that can induce RKIP and inhibit tumor growth while promoting Schwannoma cell maturation under a neurofibromatosis type 2 (NF2)-deficient condition. Nf18001 acts by selectively inhibiting the RKIP–TβR1 (transforming growth factor-β receptor 1) network, a previously established axis responsible for mediating tumor growth [131], through the degradation of the SOX2 gene. The findings strongly indicate that selective RKIP inducers could hold promise for treating both NF2 syndrome and an NF2-deficient malignant peripheral nerve sheath tumor (MPNST) [104].

Nitric oxide (NO) donors, like DETA/NO, have also shown the capability to enhance RKIP expression by inhibiting the NF-κB/YY1/Snail regulatory circuit, leading to heightened sensitivity to tumor chemo-immuno-sensitization and the inhibition of EMT and metastasis [75,132]. NPI-0052, a proteasome inhibitor, can also induce RKIP expression [108]. Inhibitors of PKCα can also prevent the phosphorylation of RKIP, activating RKIP anti-tumor activities. In fact, a few PKC inhibitors have already been used in preclinical and clinical studies, such as LXS196 (NCT02601378), Staurosporine (NCT00082017), and ruboxistaurin (NCT00133952) to name a few.

### 2.10. PD-L1 Inhibitors

In clinical settings, it has been demonstrated that immune checkpoint inhibitors (ICIs) targeting the PD-1/PD-L1 axis offer greater beneficial effects for patients with advanced or metastatic cancer compared to conventional therapies [133]. Most notably, some promising advancements include a novel PD-L1 aptamer, a short single strand of DNA that is smaller than the PD-L1 antibody, which can efficiently bind to PD-L1 with less hindrance by antigen glycosylation [134]. In addition, the role for novel small-molecule therapeutics has been an attractive target for combination with existing immune checkpoint inhibitors and/or other agents. For example, a peptide-based molecule inhibitor named AUNP-12 shows potential in mitigating immunotherapy-related adverse events (irAEs) due to its metabolic half-life [135]. Furthermore, in addition to the FDA-approved anti-PD-L1s/PD-1s mentioned earlier, there has also been a great number of emerging PD-L1 immunotherapies that have been extensively reviewed [51,136,137]. We have also collated a summary table with PD-L1 inhibitors and RKIP inducers (Table 1).

### 2.11. Combining Targeting Both RKIP and PD-L1

To date, many preclinical and clinical studies have emerged to target the PD-1/PD-L1 axis [137]. Despite ongoing effort, patients still exhibit primary resistance, failing to respond to conventional monoclonal antibodies (mAbs) targeting the PD-1/PD-L1 pathway [138]. These mAbs have also been shown to exhibit a poor penetration of tumor tissues, immune-related adverse effects, and vulnerability to drug resistance [139]. In addition, some may also develop adaptive resistance or even resistance after relapse [138]. As a result, the use of a combined therapy involving anti-PD-1/PD-L1 immunotherapy is a crucial alternative for enhanced efficacy compared to monotherapy. This leads us to consider the combination of RKIP inducers and PD-L1 inhibitors as a therapeutic approach.

As previously discussed, Nf18001 exhibits the capability to induce RKIP chemically and promote the degradation of Sox2, thereby influencing the TGF-β signaling pathway through the modulation of the RKIP–TβR1 network [104]. In parallel, the bispecific antibody (bsAb) YM101, innovatively developed by Wu et al., is designed to concurrently target PD-L1 and transforming growth factor-beta (TGF-β) [140]. TGF-β assumes a role as a negative regulator in advanced tumors, exerting suppressive effects on T cell cytotoxicity, impeding the antigen presentation of dendritic cells and fostering the differentiation of Tregs [141]. Our proposition is to investigate a combined therapeutic approach involving both Nf18001 and YM101 that arises from the potential synergistic or additive effects that may emerge from their respective mechanisms of action. Nf18001, by modulating the RKIP-TβR1 network [131], can influence TGF-β signaling, while YM101 can simultaneously target PD-L1 and TGF-β. Considering the immunosuppressive functions of TGF-β in advanced tumors, the combination of Nf18001 and YM101 holds promise as a comprehensive strategy to enhance anti-tumor immune responses and possibly promote T cell infiltration into tumor centers. Further research is warranted to elucidate and validate the potential additive effect between these drugs as well as their preclinical and clinical safeties and efficacies. The consideration of other bsAbs and RKIP inducers targeting other hyperactive pathways such as MAPK, PI3K/AKT, etc., is also worth investigating.

In addition, the consideration of natural RKIP inducers such as flavonoid Didymin with current PD-1/PD-L1 mAbs can be a cost-effective and efficient way to enhance anti-tumor responses. Didymin has already been shown to inhibit neuroblastoma growth in vivo and in vitro, induce apoptosis, and influence several cellular regulators such as vimentin, cyclin D1, and N-MYC through the upregulation of RKIP [127]. As mentioned earlier, RKIP has been shown to influence these factors through the stabilization of GSK3β signaling, subsequently affecting the stabilization of cyclin D1, N-MYC, and factors that influence EMT transition such as β-catenin, Snail, and Slug [100]. MYC and N-MYC have also been shown to exert control over PD-L1 expression in neuroblastoma, both in in vitro and in vivo settings [142]. Past studies have shown positive PD-L1 expression with Snail and vimentin *H* scores [143]. Given these connections between RKIP and PD-L1 through EMT factors and N-MYC, the combination of anti-PD-L1s and Didymin can be effective against treatments of neuroblastoma or other types of cancers. Further targeted experiments are necessary to validate these associations. Nevertheless, Didymin is non-toxic to normal tissues and has already shown benefits in the treatment of neuroblastoma and NSCLC [144]; given this, a personalized combined treatment may be worth consideration.

## 3. Clinical Trials

To date, RKIP’s clinical role has been largely unexplored, with a few trials potentially addressing RKIP upregulation indirectly by using agents that induce RKIP, namely NPI-0052, EGCG, and Rituximab. On the contrary, there has been extensive clinical trials conducted with anti-PD-L1s (atezolizumab, durvalumab, and avelumab) and PD-1s (pembrolizumab, nivolumab, and cemiplimab). Table 2 delineates clinical trials utilizing PD-L1/PD-1 mAbs alongside other interventions targeting critical pathways that are regulated by RKIP, such as MAPK or PI3K/AKT (Table 2). The first two trials (NCT03149029 and NCT02027961) utilize PD-1/PD-L1 mAbs with Dabrafenib (BRAF inhibitor) and Trametinib (ERK inhibitor) to inactivate MAPK signaling in melanoma [91]. In relation to these therapeutics, RKIP not only inhibits Raf-1 in MAPK/ERK signaling [145], but has also been shown to inhibit BRAF activity in certain cellular contexts [146], including one where RKIP downregulates BRAF activity in melanoma cell lines independent of its action with Raf-1 [147]. The fourth trial listed (NCT05253131) also targets the MAPK pathway, using MEK inhibitors Selumetinib and Bromodomain with durvalumab. The third trial combined pembrolizumab and GSK2636771 (PI3K-Beta Inhibitor) to target the PI3K signaling pathway (NCT03131908). In relation to RKIP, mechanistically, RKIP has been shown to interact with the p85 subunit of PI3K in mast cell inhibition, preventing binding of GRB2-associated binding protein 2 (Gab2), and subsequently inhibiting the activation of the PI3K/Akt/NF-κB complex [148]. In addition, RKIP can also suppress PI3K indirectly through the dysregulated NFκB/Snail/YY1/RKIP/PTEN loop mentioned earlier [54].

Additionally, we have collated available trials highlighting RKIP inducers, namely NPI-0052 and EGCG (Table 3). As mentioned earlier, NPI-0052 has been shown to induce RKIP expression in prostate cancer cell lines [108] while EGCG has been shown to induce RKIP in pancreatic adenocarcinoma [124]. In addition, both EGCG and NPI-0052 can inhibit NF-kB, potentially modulating PD-L1 expression via the PI3K/Akt pathway [149,150]. The clinical trials listed have used EGCG or NPI-0052 as therapeutic agents among other interventions in a multitude of malignant cancers, providing a basis for their safety and efficacy.

## 4. Discussion

Due to the upregulation of checkpoint ligands, the suppression of antigen presentation, irreversible CD8+ T cell exhaustion, and an immunosuppressive TME, malignant cells can escape immune surveillance and further the development of tumorigenesis [151]. In this review, we examined two crucial gene products and their expressions in the context of immune evasion: PD-L1 overexpression and RKIP under-expression. PD-L1 is a crucial immune checkpoint ligand, with inhibitory roles in CD8+ T cell effector functions. An elevated expression of PD-L1 in tumors has been reported to strongly correlate with an advanced disease state and unfavorable prognosis [40]. We have examined possible indirect and direct relationships between PD-L1 and RKIP through cross-talks with GSK3β, MAPK, and JAK-STAT signaling; the SOX2 oncogene; transcription factors YY1 and NF-κB; and cytokines IL-1β and IFN-γ. Our results have highlighted the existence of inhibitory relationships between PD-L1 and RKIP, indicating that increased RKIP levels may lead to decreased PD-L1 expression. Our bioinformatic analyses have also revealed inverse relationships between PD-L1 and RKIP in many different types of cancer. Accordingly, we have highlighted potential novel strategies to induce RKIP and downregulate the expression of PD-L1 by several interventions. We will briefly summarize these relationships below, as well as provide future perspectives into PD-L1 and RKIP at the transcriptional levels. These relationships will provide further insights and limitations for other key therapeutic approaches.

As cancer requires the constitutive expression and activation of TFs, a key aim is to deduce which TFs are actively promoting aberrant gene expression and cancer malignancy. We have established the role of RKIP with the TFs YY1 and NF-kB, allowing an indirect suppression of PD-L1 expression via PTEN and the PI3K/Akt pathway. Interestingly, one study has also indicated that YY1 can interact with the RKIP promoter region through seven distinct binding sequences, with relative binding scores ranging from 80.4% to 87.5% [152]. Given these connections, it may be possible that YY1 can regulate both RKIP and PD-L1 at the transcription levels in addition to an indirect cross-talk via the PI3K/Akt pathway. Since YY1 has the regulatory control over 7% of human genes [153], it is an extremely important node to consider for future clinical interventions.

We have also highlighted the inverse relationship between RKIP and PD-L1 via GSK-3β, influencing the canonical Wnt signaling. As reported by Al Mulla et al., RKIP has been shown to positively regulate GSK-3β kinase, and RKIP downregulation elevated the protein levels of β-catenin, Snail, SLUG, and cyclin D1 [100]. Snail has also been shown to induce PD-L1 through the Snail-driven activation of the Wnt pathway in lung cancer cells. Where, mechanistically, a drug named resveratrol can stabilize Snail, allowing the Snail inhibition of the transcription of Axin2. This destabilizes GSK-3β kinase and eventually allows accumulated β-catenin to form a complex with transcription factors like TCF/LEF and bind to the PD-L1 promoter [154]. These results also correlate with our observations of conserved motif sites for LEF1 on both RKIP and PD-L1 sequences. This indicates that Snail can positively regulate PD-L1 and inhibit RKIP expressions at the transcriptional level, possibly via the activation of Wnt signaling in order to promote the β-catenin–LEF1 complex for subsequent binding to the PD-L1 promoter. These results may also explain why the PD-L1-positive rate was much higher in patients with mesenchymal phenotypes compared to those with epithelial phenotypes [143]. Consequently, it implies that patients exhibiting mesenchymal phenotypes may be more responsive to PD-L1 immunotherapy. Thus, future considerations should explore the role of how TFs can also influence the EMT in cell plasticity and immune evasion.

Noteworthily, HIF-1 has also been reported to affect both the RKIP and PD-L1 gene products transcriptionally. PD-L1 upregulation under hypoxia was directly dependent on hypoxia-inducible factor-1α (HIF-1α)—where it increased PD-L1 expression on macrophages, dendritic cells, and tumor cells [155]. Thus, the interaction of RKIP with HIF-1α can protect against pancreatic cancer metastasis by inhibiting its hypoxia function [156]. Therefore, cancer hypoxia regulation via RKIP might be a crucial axis to target hypoxia-induced EMT and subsequent metabolic reprogramming.

We have also proposed novel strategies to combine PD-L1 mAbs with RKIP inducers. Specifically, we have proposed to combine the RKIP inducer Nf18001 and the bispecific antibody, YM101, due to similarities in TGF-β signaling. We have also proposed to combine Didymin with PD-L1 mAbs due to implications between RKIP and PD-L1 through EMT factors, GSK3β signaling, and N-MYC in treatments of neuroblastoma. Both of these dual approaches have the potential to overcome resistance mechanisms that arise from single-agent therapies and improve treatment outcomes in patients with cancer. It could also offer an additive or synergistic effect by targeting both the immune checkpoint and upstream regulators of PD-L1 expression. It is important to note that this is one such example of similar combinations, and further studies are needed to affirm the preclinical efficacy and safety of such an approach. However, given the wide variation of PD-L1 expression levels in cancers among different populations, some patients may exhibit low or even absent PD-L1 expression with their cancers [157,158]. In such cases, attempting to achieve therapeutic responses by directly reducing PD-L1 expression may prove ineffective.

Nevertheless, targeting RKIP could have broader implications beyond PD-L1 regulation. RKIP is involved in various signaling pathways, including MAPK, which is responsible for over 40% of all human cancer cases [159]. Therefore, RKIP induction strategies could have pleiotropic effects on tumor biology, making it an attractive therapeutic target for cancer treatment. If used in combination with existing PD-L1/PD-1 mAbs, this could provide an enhanced treatment approach for patients developing resistance or not responding to monotherapies. Moving forward, utilizing the therapeutic role of RKIP as an intervention is an extremely promising aspect in the field of cancer immunology.

## 5. Conclusions

The field of immunotherapies is ever expanding with ongoing efforts to implement checkpoint inhibitor agents for a subset of patients with cancer across various cancer types. However, there is still a concerning need for new therapeutic strategies to address unresponsive cases. Our investigation revealed interrelated pathways between the expression levels of PD-L1 and RKIP, where RKIP upregulation plays a role in the suppression of PD-L1 expression directly or indirectly in cancer cells. Consequently, targeting RKIP directly or modulating the associated cross-talk pathways can potentially reduce PD-L1 expression and prevent tumorigenesis. This, in turn, could render resistant malignant cells susceptible to the combined effects of checkpoint inhibitors and RKIP inducers. Therefore, this combined approach can enhance CD8+ T cell anti-tumor responses and immune surveillance, while downregulating major hyperactive pathways present in malignant cells.

## Figures and Tables

**Figure 1 cells-13-00864-f001:**
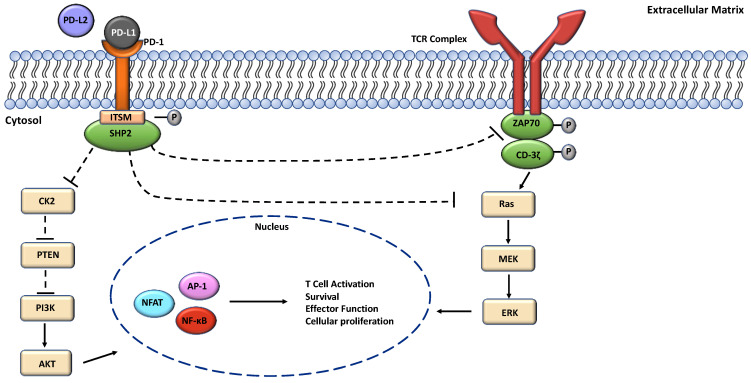
A pathway depicting the activation of the PD-1 pathway via PD-L1. Activation via PD-L1 will phosphorylate the ITSM region, triggering the recruitment of SHP2. The interactions with SHP2 and ITSM lead to the inhibition of positive signals (CD-3ζ and ZAP70) that occur through the T cell receptor (TCR). These inhibitory signals will lead to the suppression of the Ras/MEK/ERK/MAPK and the PI3K/AKT pathways. Eventually, the inhibition of those pathways will lead to the downstream suppression of transcription factors AP-1, NFAT, and NF-κB in the nucleus. These will affect T cell activation and survival, effector function, and cellular proliferation.

**Figure 2 cells-13-00864-f002:**
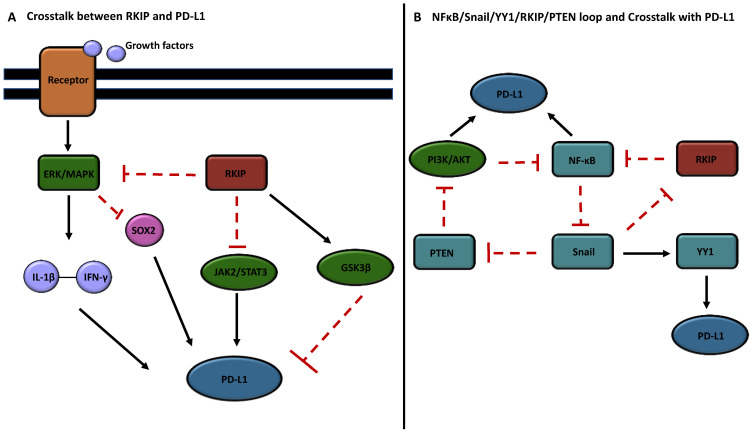
Cross-talks between RKIP and PD-L1 signaling pathways. (**A**) The pathway depicts various cross-talks between RKIP and PD-L1. The activation of the ERK/MAPK pathways lead to an increased production of cytokines IFN-γ and IL-1β, upregulating PD-L1 expression. RKIP inhibits this pathway, suppressing downstream signaling and PD-L1 expression. Moreover, RKIP inhibits SOX2 through ERK inhibition and also inhibits the JAK/STAT pathway, both of which contribute to PD-L1 upregulation. Additionally, RKIP activates GSK3β signaling, promoting PD-L1 degradation. (**B**) A pathway depicting the dysregulated loop NFκB/Snail/YY1/RKIP/PTEN. High RKIP expression inhibits NF-κB, leading to the downregulation of Snail and YY1 expressions, while promoting PTEN and downregulating the PI3K/AKT pathway, leading to reduced PD-L1 regulation. Downregulation is represented by the red blocking arrows, while upregulation is represented by the black arrows. Black arrows upregulating PD-L1 leads to tumor-inducing signaling, while the arrows mediated by RKIP seeks to show RKIP’s prevention of dysregulated signaling.

**Figure 3 cells-13-00864-f003:**
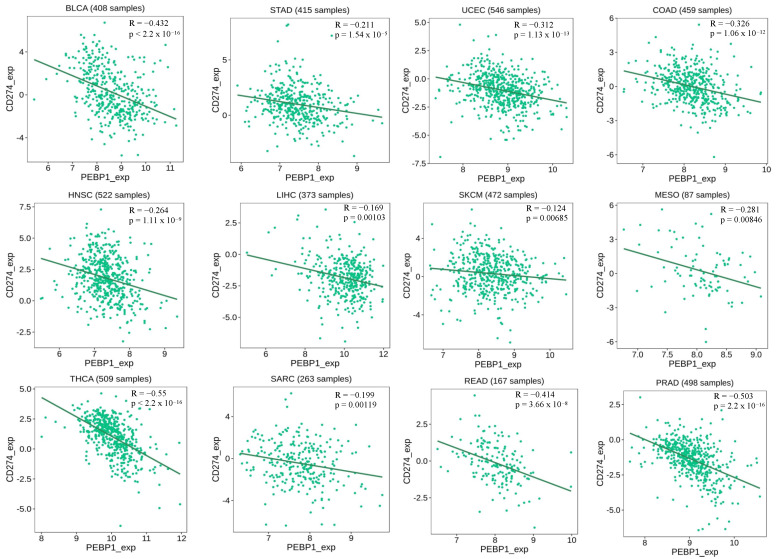
Negative correlation between RKIP expression and PD-L1 expression in different types of cancer. The graphs and data were derived and produced using TISIDE (Accessed January 2024). RKIP and PD-L1 alternative names (PEBP1/CD274) were used. The Spearman R (correlation coefficient), *p*-values, and number of samples for the individual correlations are indicated.

**Figure 4 cells-13-00864-f004:**
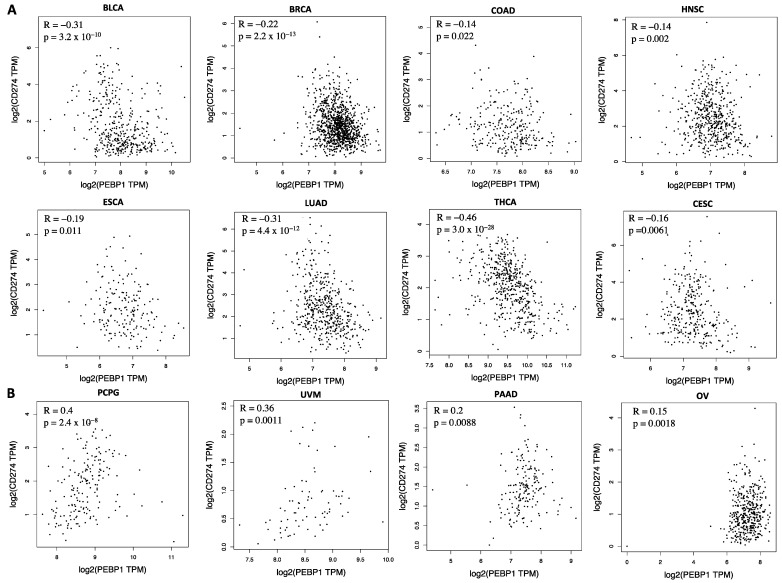
Negative and positive correlations between RKIP and PD-L1 expressions in various cancers. (**A**) (row 1–2) represents negative correlations between RKIP and PD-L1 in 8 different cancer types. (**B**) shows positive correlations between RKIP and PD-L1 in 4 different types of cancers. RKIP and PD-L1 alternative names (PEBP1/CD274) were used. Spearman correlation (R) and *p*-values are indicated in each graph. Graphs are produced using GEIPA with data from TCGA (Accessed January 2024).

**Figure 5 cells-13-00864-f005:**
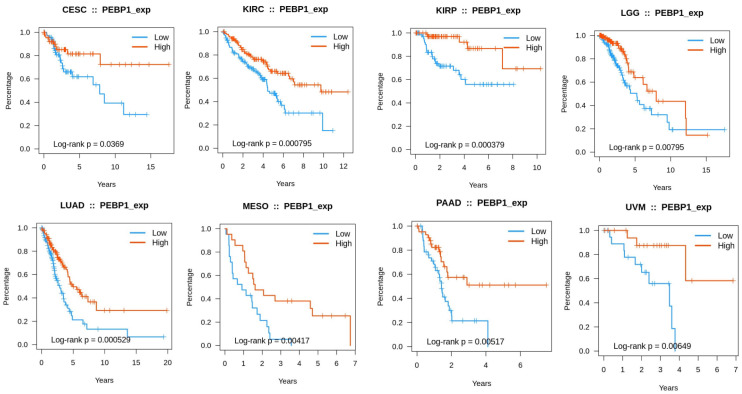
Associations between RKIP (PEBP1) expression and overall survival across human cancers. Higher expression levels of RKIP showed longer survival rates for CESC, KIRC, KIRP, LGG, LUAD, MESO, PAAD, and UVM. Graphs and data were derived and produced using TISIDE (Accessed January 2024).

**Figure 6 cells-13-00864-f006:**
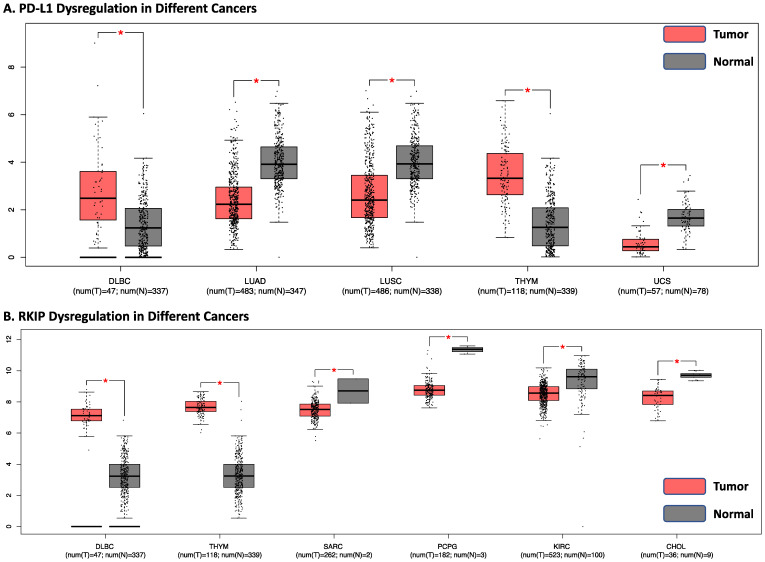
RKIP and PD-L1 gene expressions. RKIP and PD-L1 gene expression samples showing significant dysregulation (*p* < 0.01) compared to the controls. (**A**) PD-L1 dysregulation in DLBC, LUAD, LUSC, THYM, and UCS. (**A**) shows tumor expression is significantly different compared to normal tissues with PD-L1. (**B**) RKIP dysregulation in DLBC, THYM, SARC, PCPG, KIRH, and CHOL. (**B**) shows tumor expression is significantly different compared to normal tissues with RKIP. The box plots shown in pink represent tumor expression levels while the grey box plots represent expression levels in normal tissues. The relative expression levels were first log2(TPM+1)-transformed and the log2FC was defined as the median (tumor)–median (normal), where TPM is the transcript count per million. Graphs are produced using GEIPA with data from TCGA.

**Table 1 cells-13-00864-t001:** Various Agents Targeting RKIP and PD-L1. The below table lists the various agents, RKIP inducers, and PD-L1 mAbs that have been reported to target RKIP and PD-L1. Abbreviations: Epigallocatechin gallate (EGCG) and Dihydroartemisinin (DHA).

**RKIP Inducers**	**Mechanisms**	**References**
EGCG	Upregulates RKIP expression by modulating histone deacetylation. Also inhibits Snail and NF-κB activity.	[123,124,125]
Didymin	Induces apoptosis by upregulating RKIP and inhibiting N-Myc.	[127]
DHA	Induces apoptosis of cervical cancer cells via upregulation of RKIP in HeLa cells.	[128]
Nf18001	Inhibitions of the RKIP– TβR1 network.	[131]
DETA/NO	Enhances RKIP expression by inhibiting the NF-κB/YY1/Snail regulatory circuit.	[75,132]
NPI-0052	Proteasome inhibitor.	[108]
Ginseng Extract	Correlations with a significant increase in RKIP mRNA and protein expression.	[123]
**PD-L1 mAbs**	**Mechanism**	**References**
Aavelumab	Binds to PD-L1, preventing the interaction between PD-L1 and PD-1.	[50,51,52]
Atezolizumab	Binds to PD-L1, preventing the interaction between PD-L1 and PD-1.	[50,51,52]
Durvalumab	Binds to PD-L1, preventing the interaction between PD-L1 and PD-1.	[50,51,52]

**Table 2 cells-13-00864-t002:** Clinical trials. The table lists clinical trials using PD-1/PD-L1 mAbs with other interventions designed to target two major pathways: MAPK and PI3K. Refer to paper for complete descriptions of relationships with RKIP.

Type of Malignancy	Phase	Interventions	Relation to RKIP	NCT Reference
Melanoma	2	Pembrolizumab Dabrafenib Trametinib	Trial targets the MAPK pathway using ERK and BRAF inhibitors. RKIP can inhibit MAPK/ERK signaling via Raf-1 [139]. RKIP expression can also inhibit BRAF in melanoma cell line [141].	NCT03149029
Melanoma	1	Durvalumab Dabrafenib Trametinib	Trial targets the MAPK pathway using ERK and BRAF inhibitors. RKIP can inhibit MAPK/ERK signaling via Raf-1 [139]. RKIP expression can also inhibit BRAF in melanoma cell line [141].	NCT02027961
Refractory Melanoma and Other Malignant Neoplasms of Skin	1/2	GSK2636771 (PI3K-Beta Inhibitor) Pembrolizumab	Trial targets the PI3K pathway. RKIP interacts with PI3K, preventing it from binding to GRB2-associated binding protein 2 (Gab2) [142].	NCT03131908
Sarcomas Including Malignant Peripheral Nerve Sheath Tumors	1/2	Selumetinib and Bromodomain (MEK inhibitors) Durvalumab	Trial targets the MAPK pathway. RKIP inhibits MAPK and MEK signaling via Raf-1 [139].	NCT05253131

**Table 3 cells-13-00864-t003:** Clinical trials using RKIP inducers. The table lists clinical trials using RKIP inducers NPI-0052 and EGCG, with some in combination with other interventions. (Refer to text.)

Type of Malignancy	Phase	Interventions	Relation to RKIP	NCT Reference
Solid Tumors, Lymphomas, Leukemias, and Multiple Myeloma	1	NPI-0052, Dexamethasone	NPI-0052 has been shown to induce RKIP [108].	NCT00629473
Grade IV Malignant Glioma	1	NPI-0052, Radiotherapy (RT), Temozolomide (TMZ), and Optune	NPI-0052 has been shown to induce RKIP [108].	NCT02903069
Glioblastoma	3	NPI-0052, RT, TMZ	NPI-0052 has been shown to induce RKIP [108].	NCT03345095
Multiple Myeloma	2	NPI-0052	NPI-0052 has been shown to induce RKIP [108].	NCT00461045
Prostate Cancer	2	EGCG	EGCG has been shown to induce RKIP [124].	NCT00676780
Breast Cancer	2	EGCG	EGCG has been shown to induce RKIP [124].	NCT00917735
Bladder Cancer	2	EGCG	EGCG has been shown to induce RKIP [124].	NCT00666562

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
