# Peer review of "Cross-Talks between Raf Kinase Inhibitor Protein and Programmed Cell Death Ligand 1 Expressions in Cancer: Role in Immune Evasion and Therapeutic Implications"

_cells, 2024, doi:10.3390/cells13100864_

Round 1

Reviewer 1 Report

Comments and Suggestions for Authors

The authors present a comprehensive review about the individual roles and cross-talk between RKIP and PD-L1 expression in cancer. They point out the important role of PD-1/PD-L1 in the context of immune evasion and as a therapeutic target; they also describe the structure and the function of RKIP in modulating cellular growth, division and apoptosis. Moreover, they describe the interacting signaling pathways and the need for novel therapeutic approaches.  In addition to the detailed description of current research state, the authors also explored databases to investigate the interdependency of RKIP and PD-L1 and searched for ongoing clinical trials with implications for the RKIP-PD-L1 cross-talk. The review is interesting for cancer research but also for neighboring research fields such as cellular signaling and immune responses. 

Questions: 

-        Are there ideas why the correlation between RKIP and PD-L1 expression is negative in most cancer types but shows positive correlation in some types of cancer (Figure 4B)?

-        In the same direction: Is there an explanation why in some tumor types PD-L1 and RKIP expression, respectively, are upregulated while in others it is the other way around? What are the clinical implications?

-        ERK is a MAPK… in Fig. 2: why did you place MAPK downstream of ERK?

Minor: 

-        The figures describing the signaling cross-talk of RKIP and PD-L1 is complex. For the unexperienced reader it is hard to understand the context. Maybe the authors could consider stating if the up- or down-regulation of proteins (arrows vs. blocking lines) correspond to regular signaling or dysregulated (tumor-induced) signaling. 

-        Line 444: …revealed that PD-L1 was expression levels were significantly dysregulated…

-        Check citation style in line 586

-        Line 633: … expression via PTEN and the PI3K/Akt. Pathway

-        Line 642: … has been shown

Author Response

We sincerely appreciate the reviewer for his statement indicating that the review is “interesting for cancer research but also for neighboring research fields such as cellular signaling and immune responses”. We also appreciate the minor comments raised and that we are happy to respond and revise the manuscript accordingly.

1.    The reviewer was interested to discuss the findings reported in Figure 4B why some cancers did not follow the general findings in moist cancers of a negative correlation between RKIP and PD-L1. Clearly, various tumor types are heterogeneous in their makeup and signaling pathways and such alterations will influence the overall cross-talks and correlation. Another important possibility is that in some cancers the expression of RKIP is not in its active form but in its inactivated form namely phosphoRKIP. Thus, this will no longer lead to a negative correlation. The analyzes in Figure 4 A and B were based on the expression of RKIP and did not discriminate between active and inactive forms of RKIP. 

We have added the above information in the manuscript following the paragraph beginning “ Using the Gene---” line 419

2.    The reviewer also inquires about the different levels of expressions of both RKIP and PD-L1 in different tumors. It is possible that signaling pathways that trigger the cancer cells in one tumor are not of the same magnitude seen in another tumor. In such a case, the triggers for RKIP and PD-L1 will be different and lead to different levels of expressions. 
3.    The reviewer rightfully inquired why in Figure 1 and 2 MAPK was downstream of ERK? That was our error and we have corrected the figure. 
4.    The reviewer inquired about the significance of the arrows representing the upregulations and downregulations depicted in Figure 2 A and B. We have revised the legends to clearly indicate their roles (regular signaling or dysregulated signaling). 
5.    Line 444 was corrected 
6.    Line 586 (now line 601) citation number was corrected
7.    Line 646-647 was corrected to read as follows: “We have established the role of RKIP with the TFs YY1 and NF-kB, allowing the indirect inhibition of PD-L1 expression via PTEN and the PI3K/Akt pathway”.
8.    Line 642 (now line 655) was corrected to read as follows” As reported by Al Mulla et al; RKIP has been shown   to positively regulate GSK-3b kinase ----”

Reviewer 2 Report

Comments and Suggestions for Authors

The current manuscript by Ho and Bonavida provides an insight into the cross talk of two proteins involved in regulating cancer progression: namely PD-L1 (involved in cancer immune evasion) and RKIP (involved in immune activation against cancer cells). Through the review, the authors focus on different mechanisms by which PD-L1 and RKIP impact each other and are regulated separately. The manuscript provides a well described introduction for the two proteins: namely PD-L1 and RKIP, including a detailed description of their roles along with the network of proteins regulating their function. Furthermore, the authors showcase, by investigating various databases, that there exists a negative correlation between PD-L1 and RKIP expression. Finally, the review has discussed potential therapeutic strategies which involve PD-L1 inhibitors and RKIP inducers for different cancers. In addition, the reader is apprised of currently running clinical trials involving indirect upregulation of RKIP. While the manuscript encompasses several regulatory pathways and ways to target different diseases through the regulation of RKIP and PD-L1 expression, certain areas require work to improve the readability of the text. 

1. The authors should prepare a table for the chemicals inducing RKIP and inhibiting PD-L1, along with the pre-clinical studies where they have been used. This would improve the readability of the text. For example, EGCG and ginseng extract can be tabulated along with other chemicals to give a better understanding of the chemicals and their potential in treatment of different diseases. 

2. The writing is too complex at places due to the long sentences which leads to the focus deviating from the main point (example: line 61-64). It is suggested that the authors break the longer sentences (4-5 lines long) into shorter ones to convey the message in a better manner, wherever possible. 

3. The grammar and word usage are incorrect at various locations. For example, in line 397 the word “implored” is wrongly used. Line 393 has a trailing statement “Bottom of form” which does not make sense. Line 570 has a repetition of the statement “in treatments of”. 

Comments on the Quality of English Language

Dear Editor,

nicely written and interesting manuscript. Needs some editing of the language and is subseq ready to go.

Cheers Mirko Schmidt

Author Response

We thank the reviewer for the effort dedicated to read this manuscript and the important comments.
The review “requires improvement for the readability of the text “. We appreciate this comment and we have edited the manuscript and clarified areas that were not clear (highlighted)
1.    The reviewer suggested the preparation of a Table to summarize the various agents that target RKIP and PD-L1and reference the studies from which they were reported. We have prepared a Table 1 as recommended. 
2.    We agree that there were instances whereby long sentences were made and as pointed out by the reviewer of an example (lines 61-64). We have revised the full text accordingly. 
3.    The reviewer correctly points out of the grammar and word usage. We agree and we have corrected throughout the manuscript of such errors.